# *Gochujang* Ameliorates Hepatic Inflammation by Improving Dysbiosis of Gut Microbiota in High-Fat Diet-Induced Obese Mice

**DOI:** 10.3390/microorganisms11040911

**Published:** 2023-03-31

**Authors:** Eun-Ji Lee, Olivet Chiamaka Edward, Eun-Bi Seo, Eun-Gyung Mun, Su-Ji Jeong, Gwangsu Ha, Anna Han, Youn-Soo Cha

**Affiliations:** 1Department of Food Science and Human Nutrition, Jeonbuk National University, Jeonju 54896, Republic of Korea; 2Department of R & D, Microbial Institute for Fermentation Industry, Sunchang-gun 56000, Republic of Korea; 3K-Food Research Center, Jeonbuk National University, Jeonju 54896, Republic of Korea

**Keywords:** fermented food, gochujang, inflammation, gut microbiota, Korean Paradox

## Abstract

Abnormal fat accumulation with gut microbiota dysbiosis results in hepatic inflammation by upregulating the release of lipopolysaccharide (LPS) and inflammatory cytokine. *Gochujang*, a traditional fermented condiment, has beneficial effects, such as anti-colonic inflammatory effects. However, *Gochujang* has been controversial because of its high salt content (the Korean Paradox). Thus, the present study aimed to investigate the preventative effects of *Gochujang* on hepatic inflammation and related gut microbiota through discussing the Korean Paradox. The mice were divided into groups including a normal diet (ND), high-fat diet (HD), HD with salt (SALT), HD with a high percentage of beneficial microbiota *Gochujang* (HBM), and HD with diverse beneficial microbiota *Gochujang* (DBM). *Gochujang* markedly reduced lipid accumulation, hepatic injury, and inflammation response. Furthermore, *Gochujang* attenuated protein expression involved in the JNK/IκB/NF-κB pathway. Additionally, *Gochujang* regulated the gut microbiota-derived LPS production and *Firmicutes/Bacteroidetes* ratio. *Gochujang* regulated the levels of gut microbiota such as *Bacteroides, Muribaculum*, *Lactobacillus,* and *Enterorhabdus,* which were correlated with hepatic inflammation. Salt did not have foregoing effects, meaning that the salt content in *Gochujang* did not affect its anti-inflammatory effect. In conclusion, *Gochujang* showed anti-hepatic inflammation effects via reduced lipid accumulation, hepatic injury, and inflammatory response together with reorganization of gut microbiota dysbiosis regardless of salt content and the difference of micro bacteria composition.

## 1. Introduction

A high-fat diet and/or high salt diet trigger abnormal fat accumulation and dysfunction of adipose tissue, increasing the production of inflammatory cytokine and chemokine [1,2]. Then, these inflammatory factors activate intracellular signaling pathways and cause chronic hepatic inflammation [3]. Phosphorylation of c-jun N-terminal kinase (JNK), the inhibitor of kappa B (IκB), and nuclear factor kappa B (NF-κB) result in transcription of NF-κB [4,5]. As a result, the expression of pro-inflammatory cytokines, such as cyclooxygenase-2 (COX-2), is upregulated and this causes the chronic inflammatory status of hepatocytes [4,6].

Intriguingly, recent studies demonstrated that HD-induced gut microbiota dysbiosis also stimulates hepatic inflammation [7]. Dysbiosis of gut microbiota increases the production of lipopolysaccharide (LPS), elevating the release of inflammatory cytokines (e.g., tumor necrosis factor-α (TNF-α) and interleukin (IL)-1β) [8]. Therefore, many investigations have been performed to find the nutritional compounds and/or foods that improve diet-induced hepatic inflammation signaling pathways and gut microbiota dysbiosis.

As probiotics, fermented foods have various beneficial effects to health by maintaining gut health and by recovering disease-related gut microbiota dysbiosis [9]. *Gochujang*, a Korean traditional fermented food (KTFF), contains multiple microbiota, such as *Bacillus* (*B.*) *amyloliquefaciens*, *B. licheniformis*, and *B. subtilis* [9]; and it alleviates obesity, oxidative stress, and inflammatory bowel disease (IBD) [10,11,12]. Indeed, *Gochujang* improves dextran sulfate sodium (DSS)-induced IBD by reducing serum IL-1β and IL-6 levels, suppressing colonic mRNA expression of TNF-α, IL-1β, and IL-6, and reversing imbalance of gut microbiota [12]. However, the ameliorative effects of *Gochujang* on hepatic inflammation and related gut microbiota dysbiosis have not been studied. Moreover, the effects of *Gochujang* on diet-induced inflammation are also unexplored.

Many previous studies have reported diverse positive effects on health and the potential beneficial microbiota of various KTFFs, including *Gochujang* and *Ganjang* [13]. However, the effects of KTFFs have been inconclusive due to their high salt level (e.g., *Gochujang*, ~10% and *Ganjang*, 16~18%), which is strongly associated with diverse metabolic disease [10,14,15]. Paradoxically, it has been reported that Koreans have a low prevalence of hypertension and cardiovascular disease, even though Koreans consume high levels of salt from diverse KTFFs in their diet [16]. Thus, this phenomenon has been defined as the “Korean Paradox” [17]. However, the explanative research regarding the Korean Paradox has still not been adequately performed.

Therefore, this study aimed to (1) investigate the effect of *Gochujang* on hepatic inflammation and gut microbiota dysbiosis in HD-induced obese mice, (2) explore the relationship between hepatic inflammation and gut microbiota dysbiosis in HD-induced obese mice, and (3) demonstrate that the salt in *Gochujang* does not have negative effects, unlike table salt.

## 2. Materials and Methods

### 2.1. Gochujang Selection

*Gochujang* microbiota determination was conducted by Next Generation Sequencing (NGS) in the Microbial Institute for Fermentation Industry (Sunchang, Jeollabuk-do, Republic of Korea). The microbiome of 76 *Gochujang* was analyzed using next generation sequencing (NGS) analysis, and the following two *Gochujang* were selected. *Gochujang* with a high percentage of beneficial micro bacteria was named HBM *Gochujang*, and relatively few but various types of beneficial micro bacteria was named DBM *Gochujang*.

### 2.2. Material & Animal Experiment

*Gochujang* was supplied by the Microbial Institute for Fermentation Industry (Sunchang, Jeollabuk-Do, Republic of Korea). Experimental animals were male C57BL/6 mice, three-weeks-old, and were purchased from DooYeol Biotech (Seoul, Republic of Korea). The mice were housed under a 12 h/12 h light/dark cycle at 23 ± 1 °C and relative humidity at 60 ± 5%. After acclimation for three weeks, animals were randomly divided into five groups (*n* = 7): normal diet (ND, 10% fat of total kcal), high-fat diet (HD, 60% fat of total kcal), HD with 0.7% salt (SALT), HD with 10% HBM *Gochujang* (HBM), and HD with 10% DBM *Gochujang* (DBM). The compositions of the diets were calculated based on the results of the *Gochujang* compositions conducted by Eurofins Woosol (Yuseong-gu, Daejeon, Republic of Korea) (Appendix A). The mice had free access to food and water. Food intake was recorded three times a week and body weight (BW) was measured once a week. Energy efficiency (g/kcal) was calculated by body weight gain (g)/energy intake (kcal). After 13 weeks, serum and tissues were collected after anesthesia and stored at −80 °C until use. This study was approved by the Institutional Animal Care and Use Committee of Jeonbuk National University (JBNU 2021-0112).

### 2.3. Biochemistry Parameters Analysis

The levels of total cholesterol (TC), triglyceride (TG), high-density lipoprotein-cholesterol (HDL), aspartate aminotransferase (AST), and alanine aminotransferase (ALT) in serum were measured using commercial kits (Asan Pharmaceutical Co., Seoul, Republic of Korea). Very low-density lipoprotein-cholesterol (VLDL) levels were calculated using Friedewald’s formula [18]. The liver tissue in the chloroform- and methanol-mixed solution (2:1 *v*/*v*) was homogenized and centrifuged at 8000 rpm, 4 °C, for 15 min. The lower layer was collected and then evaporated at room temperature. The precipitate was dissolved at 5% Triton X-100 (Takara Blo Inc, Kusatsu, Shiga, Japan). TG and TC levels were analyzed by the commercial kit (Asan Pharmaceutical Co., Seoul, Republic of Korea). The inflammatory cytokines, including TNF-α and IL-1β, were measured using a pre-coated ELISA kit (Westlake Village, CA, USA) in accordance with the manufacturer’s instructions.

### 2.4. Micro-Computed Tomography (CT) and Histological Analysis

To analyze visceral white adipose tissue (VAT) volume, Micro-CT was conducted using a high-resolution in vivo micro-CT system (Skyscan 1076, Konitch, Belgium) at the Center for University-wide Research Facilities (CURF) at Jeonbuk National University. Fat volume was found by CTAn software (Skyscan Co., Konitch, Belgium). Liver tissue was fixed with 10% formalin for 48 h and further processed by the KP&T Company (Cheongju, Chungcheongbuk-do, Republic of Korea). Hematoxylin and eosin (H&E) staining was performed. Stained samples were analyzed by optical microscopy (DM2500, Leica, Germany) in the CURF of Jeonbuk National University. Image-J software (US National Institutes of Health, Bethesda, MD, USA) was used to determine the quantification of the hepatic lipid content.

### 2.5. Quantitation Real-Time PCR (qRT-PCR)

RNA was extracted from liver tissue using the TRIzol reagent (Takara Korea, Seoul, Republic of Korea). cDNA was synthesized with the PrimeScript RT Master Mix (Takara, Kyoto, Japan). qRT-PCR analyses were performed with SYBR Green PCR Master MIX (Biosystems, Woolston, Warrington, UK) and real-time PCR (7500 Real-Time PCR System, Foster City, CA, USA). Relative quantification of gene expression calculated relative to β-actin was used as a housekeeping gene, and the relative quantification of gene expression was calculated using the 2^∆∆Ct^ method. The primer sequences are listed in Appendix A.

### 2.6. Western Blot

Liver tissue was homogenized in RIPA lysis buffer (Biosesang, Seoul, Republic of Korea) containing a protease inhibitor and phosphatase inhibitor cocktail (EMD Millipore Corp., Burlington, MA, USA), and was centrifuged (4 °C, 12,000× *g*, 15 min) following the supernatant that was collected. Proteins were electrophoresed on 10–12% SDS-polyacrylamide gels and transferred to polyvinylidene difluoride membranes (Bio-Rad Laboratories, Hercules, CA, USA) followed by blotting with primary antibodies (1:1000); p-IκB, p-p65, COX-2 (Santa Cruz, Dallas, TX, USA), IκB, p65, p-SAPK/JNK, SAPK/JNK, and β-actin (Cell Signaling Technology, Danvers, MA, USA). The membranes were washed and probed with horseradish-peroxidase-conjugated anti-rabbit or anti-mouse secondary antibodies (1:2000).

### 2.7. Gut Microbiota Analysis

At the end of the intervention period, fresh feces were collected and stored at −80 °C and processed further by the Microbial Institute for Fermentation Industry (Sunchang, Jeollabuk-Do, Republic of Korea). Microbiome composition, alpha-diversity, and beta-diversity were analyzed through the EzbioCloud 16S-based Microbiome Taxonomic Profiling software (ChunLab, Inc., Seoul, Republic of Korea). The levels of endotoxins in mouse serum were measured by using a pierce chromogenic endotoxin quantitation kit (Thermo Fisher Scientific Inc., Rockford, IL, USA), according to the manufacturer’s manual.

### 2.8. Statistical Analysis

All results are expressed as mean ± standard error of means (SEM). Analysis was performed using SPSS version 22.0 (SPSS Institute, Chicago, IL, USA) by one-way ANOVA, followed by Duncan’s Multiple Range Test (a > b), and the criterion for statistical significance was set at *p* < 0.05. In the case that comparison between the two groups was required, the *t*-test was used. * *p* < 0.05, ** *p* < 0.01, and *** *p* < 0.001 were considered to be statistically significant.

## 3. Results

### 3.1. Microbial Characteristics of HBM and DBM Gochujang

We confirmed the differences between HBM and DBM *Gochujang* by analyzing beneficial bacteria percentage and composition. OTUs, ACE, CHAO, Simpson, and Phylogenetic diversity in HBM *Gochujang* were higher than those in DMB *Gochujang*, but the Shannon index was lower in HBM *Gochujang* compared with DBM *Gochujang* (Table 1). The percentage of beneficial micro bacteria in HBM *Gochujang* (94.32%) was higher than those in DBM *Gochujang* (11.41%) (Figure 1). However, HBM *Gochujang* was dominated by *Bacillus subtilis* group, while DBM *Gochujang* had various beneficial micro bacteria, such as *Bacillus subtilis* group, *Enterococcus faecium* group, and *Pediococcus acidilactici* group (Figure 1).

### 3.2. Gochujang Ameliorates Obesity and Hyperlipidemia in HD-Induced Obese Mice

As described in Figure 2A, the HD and SALT groups showed a significant increase in body weight compared with the ND group, whereas both *Gochujang* groups showed a significantly lower body weight than the HD and SALT groups. There was no significant difference in food intake among the HD groups (Figure 2B). However, both the HD and SALT groups showed a higher feed efficiency and energy efficiency than both *Gochujang* groups, suggesting that the weight gain of these groups was not coming from food intake (Figure 2C,D). Liver and spleen weights of the HD and SALT groups were significantly higher than the ND group, whereas those of both *Gochujang* groups were markedly lower compared with the HD and SALT groups (Figure 2E,F).

Compared to the ND group, the alternative and relative visceral fat volumes of the HD and SALT groups were notably higher, but both *Gochujang* groups had a significantly reduced visceral fat volume compared with the HD and SALT groups (Figure 3A–C). Consistently, the alternative and relative epididymal fat weight also significantly lowered in the HBM and DBM groups compared with the HD and SALT groups (Figure 3D,E). Compared to the ND groups, the HD and SALT groups had increased levels of serum TG, TC, and VLDL, while the HBM and DBM groups had a significantly reduced level of those parameters when compared with the HD and SALT groups (Table 2). These observations imply that *Gochujang* improves HD-induced obesity by improving fat accumulation and serum dyslipidemia.

### 3.3. Gochujang Alleviates Hepatic Injury, Lipid Accumulation, and Inflammation in HD-Induced Obese Mice

The elevations of the enzymatic activity of serums AST and ALT are associated with hepatic injury [19]. Additionally, increased hepatic lipid accumulation results in hepatic inflammation [20]. Thus, AST and ALT serum levels and hepatic lipid levels were measured. Compared with the ND group, the ALT level of the HD and SALT groups were higher with and/or without statistical significance, while the HBM and DBM groups showed a significant reduction of AST and ALT levels compared with the HD and SALT groups (Figure 4A,B). Hepatic TG and TC levels of the HD and SALT groups were significantly higher than the ND group, but those of both *Gochujang* groups were notably lower than in the SALT and HD groups (Table 3). Furthermore, hepatic lipid droplets in the HD and SALT groups were higher than in the ND group; however, the HBM and DBM groups were dramatically decreased compared with the HD and SALT groups (Figure 4C,D). Additionally, the HD and SALT groups showed leukocytic infiltration, while other groups did not (Figure 4C).

To further investigate the role of *Gochujang* in hepatic inflammation amelioration, hepatic pro-inflammatory cytokine levels were analyzed. As expected, the serum cytokine levels of IL-1β and TNF-α in the HD and SALT groups were significantly increased compared with the ND group, while the HBM and DBM groups showed significantly lower levels of those of the SALT group (Figure 4E). The hepatic mRNA expression levels of IL-1β, TNF-α, NF-κB, and COX-2 in the HD and SALT groups were notably elevated compared with the ND group, while both *Gochujang* groups notably decreased when compared with the SALT and HD groups (Figure 4F). These results indicate that *Gochujang* counteracts HD-induced hepatic injury, steatosis, and inflammatory response.

### 3.4. Gochujang Inhibits Activation of JNK/IκB/NF-κB Pathway in HD-Induced Obese Mice

Since activation of JNK, IκB, and NF-κB results in hepatic inflammation and steatohepatitis [21,22,23], phosphorylation of those proteins was probed to evaluate the inhibitory effects of *Gochujang* in HD-induced hepatic inflammation. The phosphorylation levels of JNK, IκB, and NF-κB were notably higher in the HD and SALT groups compared with the ND group (Figure 5). Compared with the SALT group, phosphorylation of those proteins in the HBM and DBM groups were decreased with and/or without statistical significance. Expression of COX-2 in the HD and SALT groups had a slight increased tendency compared with the ND group, while both *Gochujang* groups showed a decreased tendency of COX-2 expression compared with the SALT group. These data suggested that activation of the JNK/IκB/NF-κB pathway is involved in HD-induced inflammatory response; *Gochujang* suppresses signaling pathway activation to exert its anti-inflammatory effects.

### 3.5. Gochujang Changes the Gut Microbiota Composition in HD-Induced Obese Mice

Dysbiosis of gut microbiota by HD is strongly related to hepatic inflammation [24]. To investigate gut microbiota dysbiosis in HD-induced obese mice and *Gochujang*’s effects on it, gut microbiota composition and bacteria-derived LPS levels were analyzed. The results of alpha analyses including ACE, Chao1, Shannon entropy, Simpson index, and Phylogenetic diversity indicated that both *Gochujang* groups, HBM and DBM, increased the microbial species’ richness compared with the HD and SALT groups, while the HD and SALT groups’ parameters were lowered and/or unchanged compared with the ND group (Figure 6A–E). We confirmed that, through 3D-principal component analysis (PCoA), the samples were separated into ND- and HD-fed mice based on PC1, and the HD and SALT groups were separated from the Gochujang groups based on PC1 (Figure 6F). At species level, the HD and SALT groups clustered together, while the Gochujang groups clustered with the ND group. Additionally, these alterations led to changes in the bacteria-derived LPS levels. Compared with the ND group, HD and SALT groups showed markedly elevated levels of LPS, whereas both *Gochujang* groups had significantly lowered levels of LPS compared with the HD and SALT groups (Figure 6G).

Next, the detailed alterations of gut microbiota community were further analyzed at the phylum and genus levels. At the phylum and genus levels, all groups exhibited different bacterial community structures. At the phylum level, *Firmicutes* and *Bacteroidetes* were dominant in all groups (Figure 7A). The *Firmicutes*-to-*Bacteroidetes* ratio (F/B ratio) tended to decrease in the HBM and DBM groups compared with the HD and SALT groups with and/or without statistical significance, while the HD and SALT groups showed significantly increased ratios compared with the ND group (Figure 7B). At the genus level, gut microbiota communities were either upregulated or downregulated based on the groups (Figure 8A). The levels of *Bacteroides*, *Anaerotruncus,* and *Muribaculum* were significantly downregulated in the HD and SALT groups compared with the ND group; however, those microbiota were significantly upregulated in the HBM and DBM groups (Figure 8B). On the other hand, the HD and SALT groups had significantly higher levels of *Lactobacillus*, *Clostridium*, and *Enterorhabdus* than the ND group, whereas those microbiota were markedly decreased in both *Gochujang* groups (Figure 8C). These observations suggest that *Gochujang* restores HD-induced gut microbiota dysbiosis and restructures gut microbiota communities.

### 3.6. Correlation between Hepatic Inflammation and Key Gut Microbiota

To illustrate correlations between key physiological markers of hepatic inflammation and gut microbiota alterations at the genus level, Spearman’s correlation coefficient was analyzed (Figure 9). *Akkermansia*, *Bacteroides*, *Anaerotruncus*, *Emergencia*, and *Turicibacter* were negatively correlated with body weight and TG and TC levels, while *Acetatifactor*, *Clostridium, Lactobacillus*, *Olsenella*, *Paeniclostridium*, and *Proteus* were positively correlated with those parameters. *Acutalibacter*, *Agathobaculum*, *Muribaculum*, *Oscillibacter*, *Paludicola*, *Parvibacter*, *Prevotella,* and *Turicibacter* were negatively correlated with ALT and AST serum levels. Furthermore, negative correlations were identified between expression of the JNK/IκB/NF-κB pathway and/or *Bacteroides*, *Muribaculum,* and *Turicibacter* while *Acetatifactor* was positively correlated with p-JNK and NF-κB. *Bacteroidetes*, *Alloprerovotela*, *Alistipes*, and *Anaerotruncus* were negatively correlated with COX-2, IL-1β, and TNF-α, however, *Enterorhabdus*, *Lactobacillus*, and *Clostridium* were positively correlated with these parameters. Importantly, *Anaerotruncus*, *Bacteroides*, *Muribaculum*, *Lactobacillus*, *Clostridium,* and *Enterorhabdus* were closely connected with obesity as well as hepatic lipid accumulation and inflammation parameters.

## 4. Discussion

Persistent HD intake results in the activation of a pro-inflammatory response and imbalance of gut microbiota, leading to chronic inflammation [25,26,27]. As one of the commonly consumed KTFFs, *Gochujang* exerts multiple health benefits, including anti-inflammatory effects [28]. For example, *Gochujang* ameliorates colonic inflammation by suppressing TNF-α and IL-6 gene expressions and by recovering gut microbiota dysbiosis in DSS-derived IBD [12]. Regardless of many earlier findings, the positive functions of *Gochujang* have been controversial because of its high salt concentration. The present study found that *Gochujang* reduced hepatic inflammation by reducing fat accumulation, hyperlipidemia, inflammatory parameters, and JNK/IκB/NF-κB pathway activation (Figure 10). Moreover, HD-induced gut microbiota imbalance was also improved by *Gochujang*, and these alterations of gut microbiota were correlated with the improvement of hepatic inflammation. Lastly, the high salt level of *Gochujang* did not have an impact on those advantageous outcomes.

Increased fat accumulation and lipid indicators upregulate inflammatory response and activate related signaling pathways [1]. Consistent with previous observations [10], the current study found that *Gochujang* reduced body weight, visceral fat volume, epididymal fat weight, and lipid levels in both the serums and livers of HD-induced obese mice. Furthermore, *Gochujang* strongly suppressed hepatic inflammatory cytokine levels, mRNA expression of inflammatory markers, and phosphorylation of JNK/IκB/NF-κB pathways. Indeed, *Gochujang* significantly decreased phosphorylation of JNK and IκB, but phosphorylation of NF-κB and expression of COX-2 protein were slightly reduced in HD-induced obese mice. As a key controller of COX-2 production, NF-κB plays an important role in the occurrences of inflammatory responses by translocating into the nucleus [21,22]. Therefore, future studies are needed to investigate *Gochujang*’s effect on NF-κB translocation to establish detailed underlying molecular mechanisms of *Gochujang*’s inhibitory effects on hepatic inflammation.

Recently, the roles of gut microbiota in the subject’s health and disease have been highlighted [29,30]. Importantly, gut microbiota play a role in the development of hepatic inflammation; thus, understanding the gut-liver axis in hepatic inflammation is important to improve the prevalence of inflammation-related liver disease, including steatohepatitis [31]. The present study demonstrated that *Gochujang* changed the diversity of gut microbiota and its composition; moreover, it decreased the gut microbiota-derived LPS level and the F/B ratio. According to previous studies, the *Muribaculum* genera is closely related to HD-induced obesity and hepatic steatosis and the *Turicibacter* genera is associated with liver fibrosis and non-alcohol fatty liver disease (NAFLD) [32,33]. These genera were significantly increased by *Gochujang* and had negative correlations with hepatic inflammatory and injury parameters, and lipid accumulation and JNK/IκB/NF-κB pathways, respectively. Consistent with previous studies, *Lactobacillus* levels increased when HD were consumed, but *Gochujang* showed a decreased *Lactobacillus* level similar to that of ND-fed mice [34]. This outcome might be because *Gochujang* alleviated gut dysbiosis due to HD and salt diets. Hence, the understanding of the exact functions of *Lactobacillus* in the association between gut microbiota and obesity-induced hepatic inflammation are essentially required in future studies.

The Korean Paradox proposes the possibility of the different effects of salt in fermented foods and the additive table salt [28]. Indeed, *Gochujang*-fed mice experienced improvements in DSS-induced IBD, while additive table salt-fed mice (same amount of salt in *Gochujang*) did not show any amelioration [12]. Furthermore, *Doenjang*, another signature KTFF, strongly improves high-salt-diet-induced blood pressure, regardless of its high level of salt [35]. The current study also observed that *Gochujang* showed a strong anti-inflammatory effect in hepatic inflammation throughout the above underlying mechanisms, while the supplementation of salt with HD accelerates hepatic inflammation and gut microbiota imbalance. Therefore, in addition to earlier findings, the present study also supports the distinct roles between additive salt and salt in KTFFs, allowing for critical future study directions, and including the understanding of different salt metabolisms between the salt in fermented foods and additive salt.

In conclusion, the present study demonstrated that *Gochujang* ameliorates hepatic inflammation by decreasing fat accumulation, dyslipidemia in serum and liver, production of pro-inflammatory parameters, and phosphorylation of the JNK/IκB/NF-κB pathway, but also by reshaping gut microbial composition in HD-induced obese mice. Moreover, *Gochujang* exerts those effects regardless of its high salt level. These data provided a scientific basis for understanding the effects of *Gochujang* on the interactions between gut microbiota and hepatic inflammation, and the Korean Paradox, regardless of its difference of micro bacteria composition.

## Figures and Tables

**Figure 1 microorganisms-11-00911-f001:**
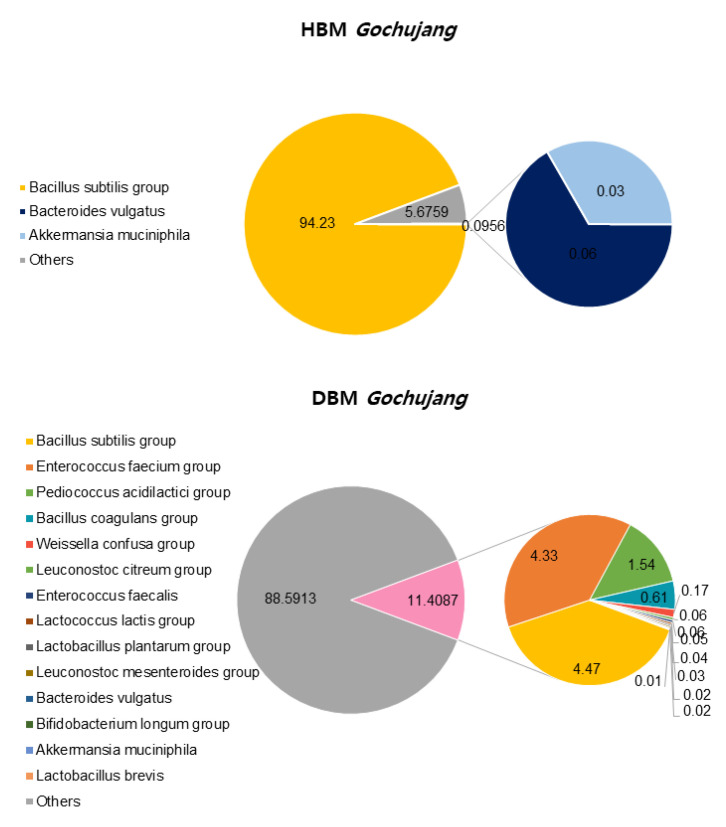
Beneficial bacteria composition of HBM and DBM *Gochujang*.

**Figure 2 microorganisms-11-00911-f002:**
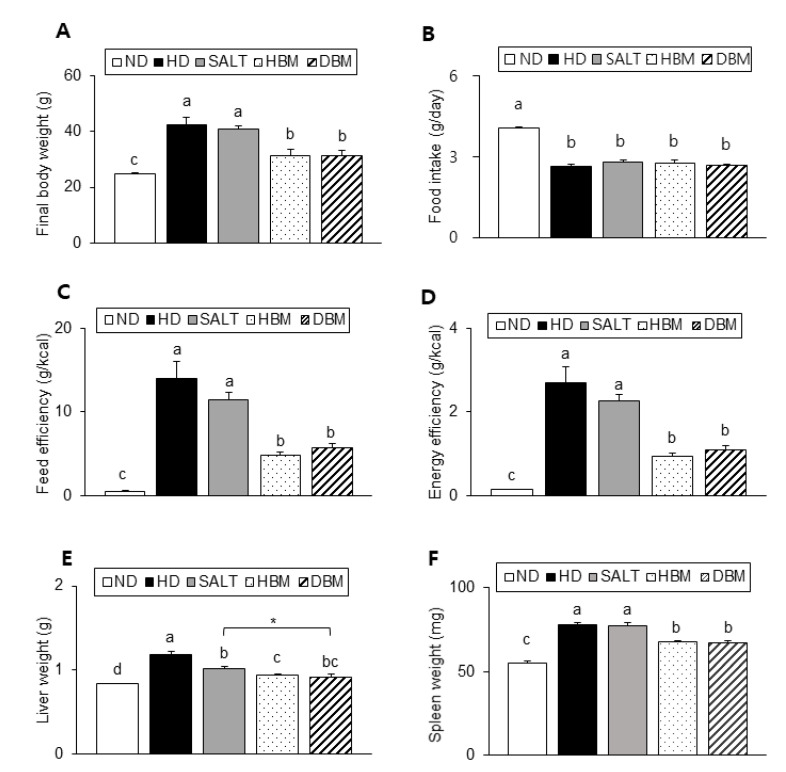
Effects of *Gochujang* on body and organs weight in the HD-fed mice. (**A**) Final body weight of mice; (**B**) Food intake; (**C**) Feed efficiency; (**D**) Energy efficiency; (**E**) Liver weight; (**F**) Spleen weight. Values are mean ± SEM. a–d means with the different letters on the bar (*p* < 0.05). (a > b > c > d). ***** *p* < 0.05. ND, normal diet group; HD, high-fat diet group; SALT, HD with 0.7% salt group; HBM, HD with 10% of HBM *Gochujang* group; DBM, HD with 10% of DBM *Gochujang* group.

**Figure 3 microorganisms-11-00911-f003:**
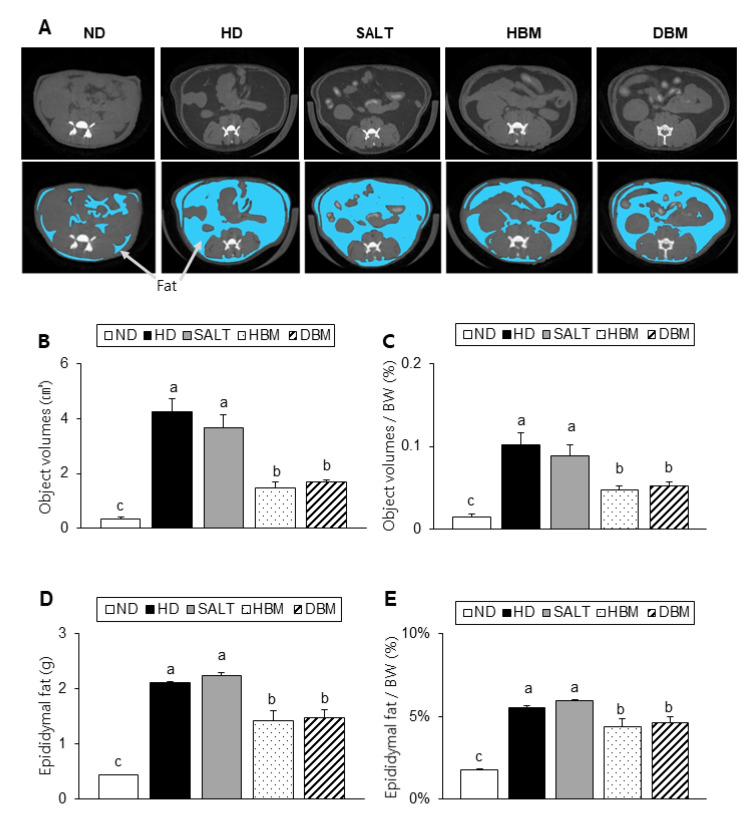
Effects of *Gochujang* on dyslipidemia in the HD-fed mice. (**A**) Micro-CT image of abdominal area (dark gray portions indicate fat tissue, which illustrated under picture with blue portions, white portions indicate bone, and light gray portions indicate organs); (**B**) Fat tissue volumes at the micro-CT; (**C**) Fat volume at the micro-CT per body weight (BW); (**D**) Epididymal fat; (**E**) Epididymal fat per BW. Values are mean ± SEM. a–c means with the different letters on the bar (*p* < 0.05). (a > b > c). ND, normal diet group; HD, high-fat diet group; SALT, HD with 0.7% salt group; HBM, HD with 10% of HBM *Gochujang* group; DBM, HD with 10% of DBM *Gochujang* group.

**Figure 4 microorganisms-11-00911-f004:**
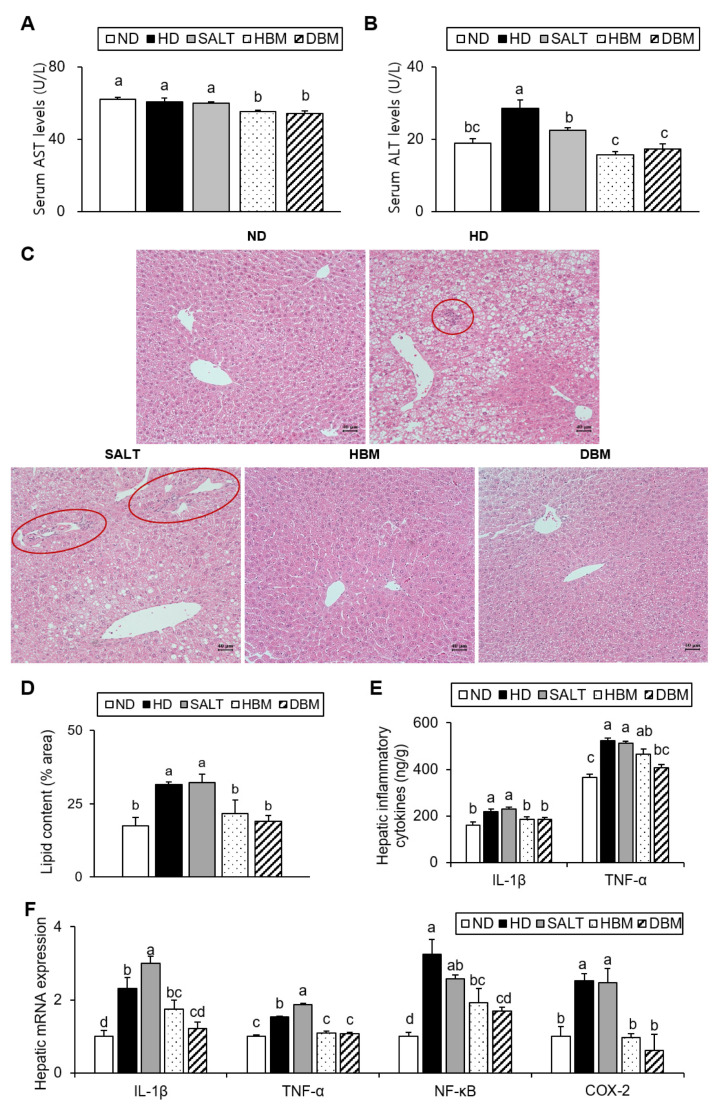
Effects of *Gochujang* on hepatic lipid accumulation and inflammation in the HD-fed mice. (**A**) Serum AST level; (**B**) Serum ALT level; (**C**) Histological analysis (200×) of liver tissues by H&E staining (red circle area showed periportal mononuclear leukocytic infiltration); (**D**) quantitation of liver lipid content shown as percentage of the total histological area; (**E**) Hepatic levels of IL-1β and TNF-α; (**F**) Relative mRNA expression of IL-1β, TNF-α, NF-kB, COX-2 in liver tissue. Values are mean ± SEM. a–d means with the different letters on the bar (*p* < 0.05). (a > b > c > d). ND, normal diet group; HD, high-fat diet group; SALT, HD with 0.7% salt group; HBM, HD with 10% of HBM *Gochujang* group; DBM, HD with 10% of DBM *Gochujang* group; AST, Aspartate aminotransferase; ALT, Alanine aminotransferase; IL, interleukin; TNF-α, tumor necrosis factor alpha; NF-κB, nuclear factor kappa B; COX-2, cyclooxygenase-2.

**Figure 5 microorganisms-11-00911-f005:**
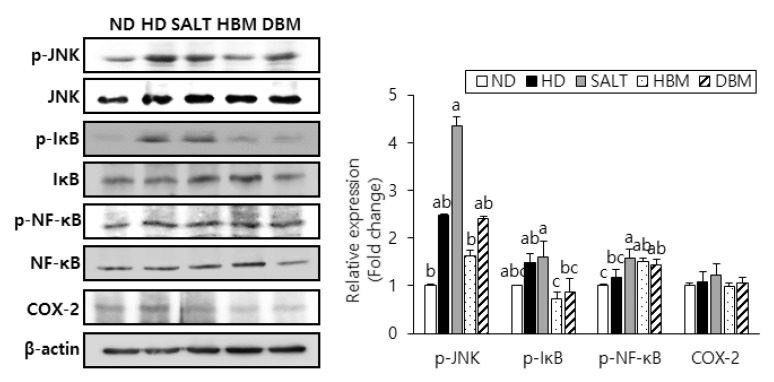
Effects of *Gochujang* on protein expression of JNK/NFκB pathway and hepatic inflammation in the HD-fed mice. Hepatic p-JNK, p-IκBα, p-NF-κB (p-p65), and COX-2 in mice. Values are mean ± SEM. a–c means with the different letters on the bar (*p* < 0.05). (a > b > c). ND, normal diet group; HD, high-fat diet group; SALT, HD with 0.7% salt group; HBM, HD with 10% of HBM *Gochujang* group; DBM, HD with 10% of DBM *Gochujang* group; p-, phospho-; IκB, inhibitor of κB; NF-κB, nuclear factor kappa B; COX-2, cyclooxygenase-2.

**Figure 6 microorganisms-11-00911-f006:**
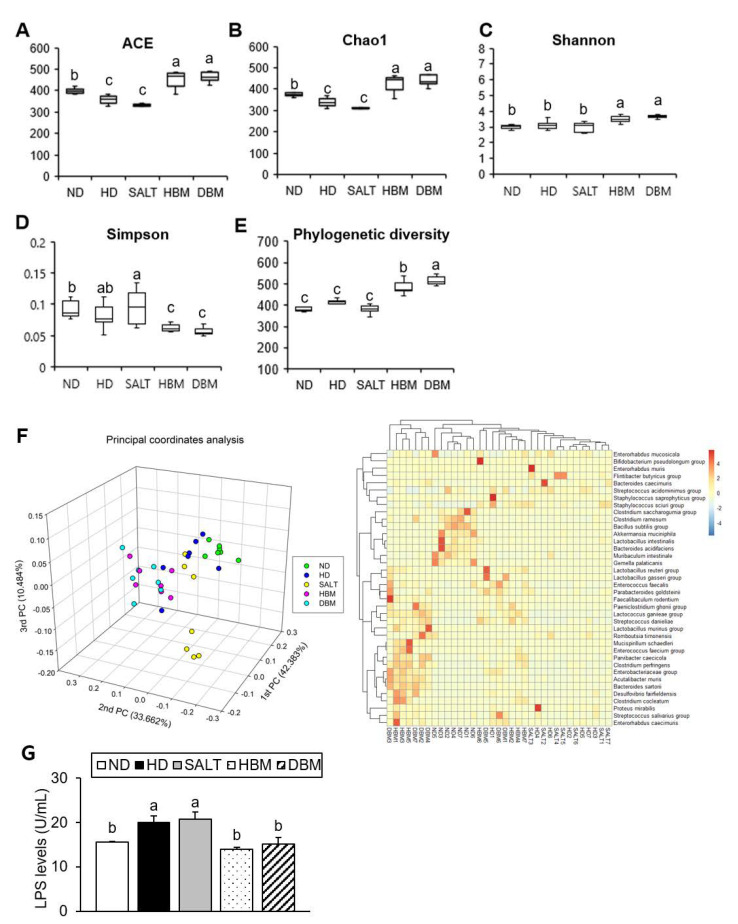
Effect of *Gochujang* on the relative abundance of gut microbial community in the HD-fed mice. (**A**) ACE; (**B**) Chao1; (**C**) Shannon entropy; (**D**) Simpson index; (**E**) Phylogenetic diversity; (**F**) 3D-Principal coordinate analysis (PCoA) and Heat map at species level; (**G**) LPS levels. Values are mean ± SEM. a–c means with the different letters on the bar (*p* < 0.05). (a > b > c). ND, normal diet group; HD, high-fat diet group; SALT, HD with 0.7% salt group; HBM, HD with 10% of HBM *Gochujang* group; DBM, HD with 10% of DBM *Gochujang* group.

**Figure 7 microorganisms-11-00911-f007:**
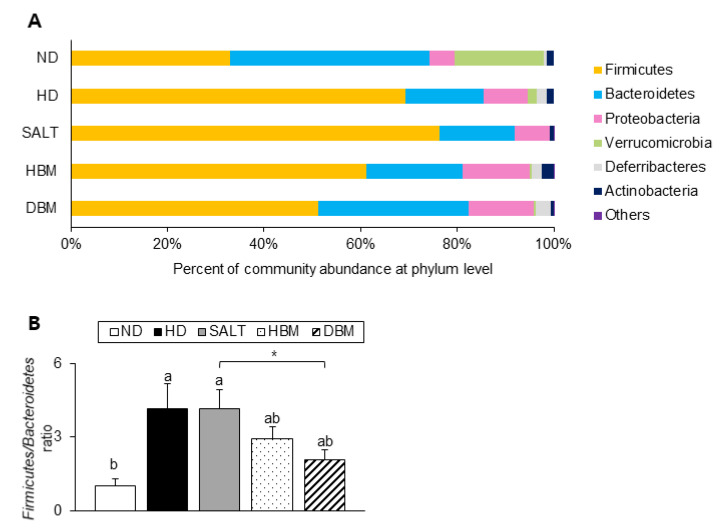
Effect of *Gochujang* on the population structure of gut microbiota at phylum level in the HD-fed mice. (**A**) Microbial community at the phylum level in mice; (**B**) *Firmicutes*-to *Bacteroidetes* ratio in mice. Values are mean ± SEM. a–b means with the different letters on the bar (*p* < 0.05). (a > b). * *p* < 0.05. ND, normal diet group; HD, high-fat diet group; SALT, HD with 0.7% salt group; HBM, HD with 10% of HBM *Gochujang* group; DBM, HD with 10% of DBM *Gochujang* group.

**Figure 8 microorganisms-11-00911-f008:**
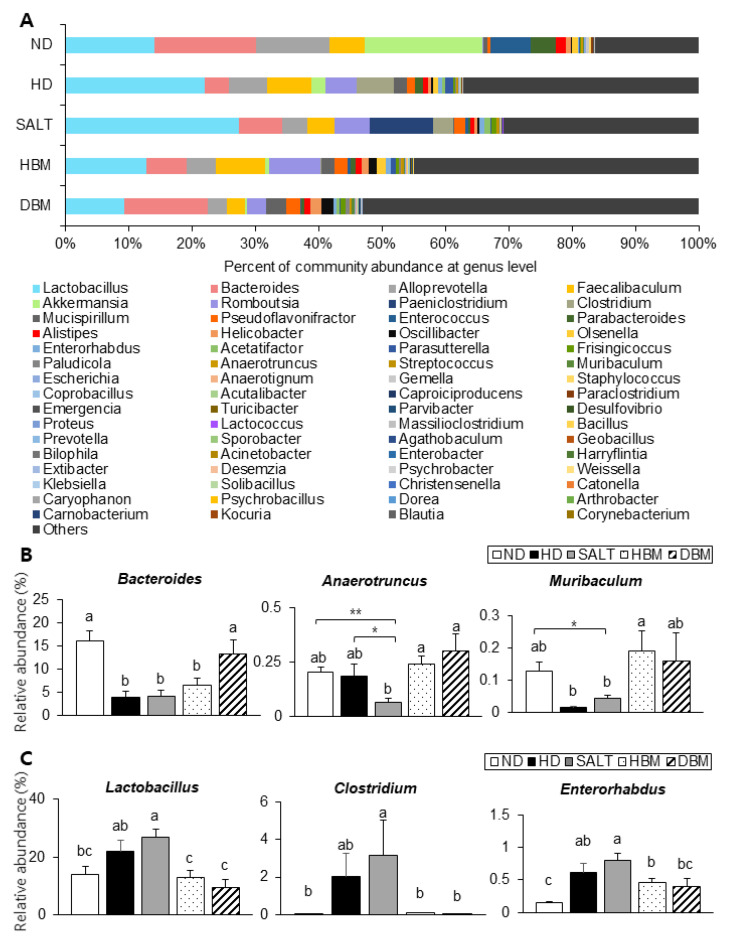
Effect of *Gochujang* on the population structure of gut microbiota at genus level in the HD-fed mice. (**A**) Microbial community at the genus level in mice; The relative abundance of (**B**) *Bacteroides*, *Muribaculum*, *Anaerotruncus*; (**C**) *Lactobacillus*, *Enterorhabdus*, and *Clostridium* at genus level. Values are mean ± SEM. a–c means with the different letters on the bar (*p* < 0.05). (a > b > c). * *p* < 0.05 and ** *p* < 0.01. ND, normal diet group; HD, high-fat diet group; SALT, HD with 0.7% salt group; HBM, HD with 10% of HBM *Gochujang* group; DBM, HD with 10% of DBM *Gochujang* group.

**Figure 9 microorganisms-11-00911-f009:**
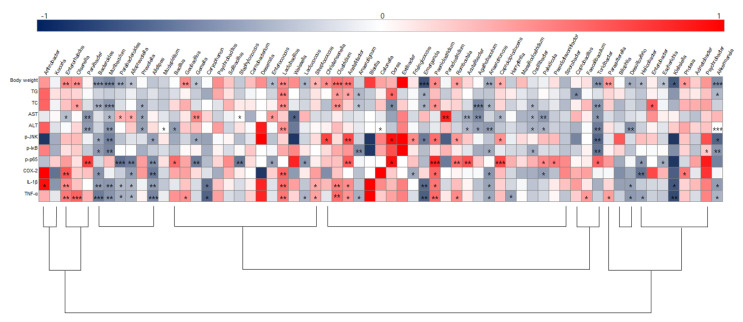
Heatmap of spearman correlation coefficient between key intestinal microbiota and hepatic inflammation traits. The intensity of the color represents the degree of association (dark blue, negative correlation; red, positive correlation). * *p* < 0.05, ** *p* < 0.01, and *** *p* < 0.001. ND, normal diet group; HD, high-fat diet group; SALT, HD with 0.7% salt group; HBM, HD with 10% of HBM *Gochujang* group; DBM, HD with 10% of DBM *Gochujang* group.

**Figure 10 microorganisms-11-00911-f010:**
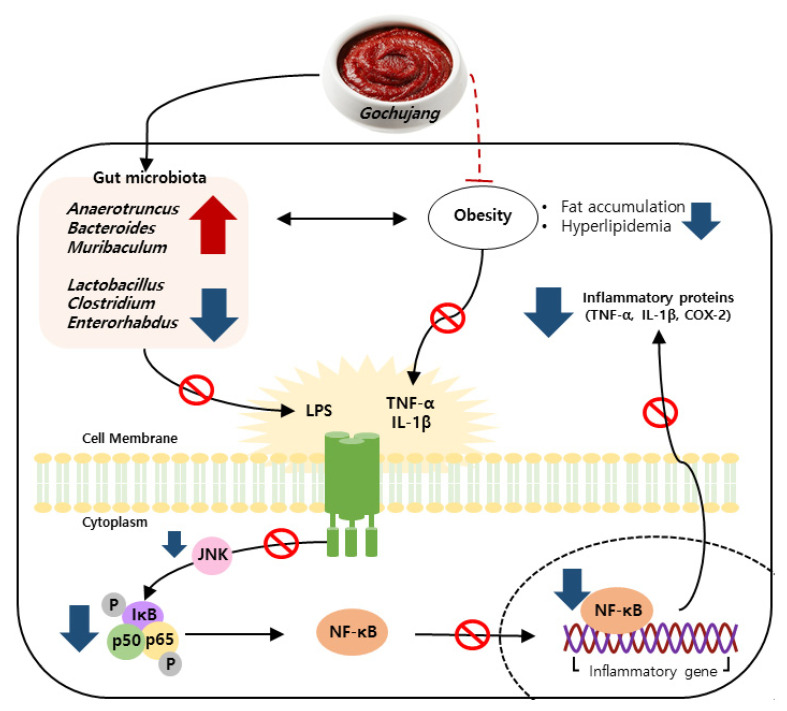
Schematic depiction of the anti-hepatic inflammatory mechanism of *Gochujang* in HD induced obese mice.

**Table 1 microorganisms-11-00911-t001:** α-diversity of *Gochujang*.

	OTUs	ACE	CHAO	Shannon	Simpson	Phylogenetic Diversity
HBM	460	504.29	499.16	0.50	0.89	1051
DBM	128	136.91	135.50	1.47	0.47	338

**Table 2 microorganisms-11-00911-t002:** Serum lipid levels.

Parameters	ND	HD	SALT	HBM	DBM
TG (mg/dL)	35.1 ± 1.3 ^d^	65.7 ± 1.0 ^a^	63.3 ± 1.4 ^a^	41.6 ± 1.6 ^c^	49.1 ± 1.5 ^b^
TC (mg/dL)	64.4 ± 1.7 ^c^	89.2 ± 0.9 ^a^	83.3 ± 0.9 ^a^	75.1 ± 1.7 ^b^	75.6 ± 4.6 ^b^
LDL (mg/dL)	11.3 ± 0.8 ^b^	21.0 ± 2.4 ^a^	18.4 ± 0.9 ^ab^	13.1 ± 2.6 ^b^	14.5 ± 3.7 ^ab^
VLDL (mg/dL)	7.0 ± 0.3 ^d^	13.2 ± 0.2 ^a^	12.7 ± 0.3 ^a^	8.3 ± 0.3 ^ab^	9.8 ± 0.3 ^b^
HDL (mg/dL)	45.3 ± 0.8 ^a^	58.7 ± 2.5 ^c^	54.5 ± 1.2 ^ab^	52.7 ± 1.3 ^c^	50.7 ± 2.6 ^b^

Values are mean ± SEM. a–d means with the different letters (*p* < 0.05). (a > b > c > d). ND, normal diet group; HD, high-fat diet group; SALT, HD with 0.7% salt group; HBM, HD with 10% of HBM Gochujang group; DBM, HD with 10% of DBM Gochujang group; TG, triglyceride; TC, total cholesterol; LDL, low-density lipoprotein-cholesterol; VLDL, very low-density lipoprotein-cholesterol; HDL, high-density lipoprotein-cholesterol.

**Table 3 microorganisms-11-00911-t003:** Hepatic lipid levels.

Parameters	ND	HD	SALT	HBM	DBM
TG (mg/g liver)	1.0 ± 0.9 ^b^	2.6 ± 1 ^ab^	3.1 ± 1.7 ^a^	1.5 ± 1.0 ^b^	1.8 ± 1.3 ^ab^
TC (mg/g liver)	1.8 ± 0.5 ^c^	2.3 ± 0.3 ^ab^	2.4 ± 0.6 ^a^	1.9 ± 0.3 ^bc^	1.6 ± 0.2 ^c^

Values are mean ± SEM. a–c means with the different letters (*p* < 0.05). (a > b > c). ND, normal diet group; HD, high-fat diet group; SALT, HD with 0.7% salt group; HBM, HD with 10% of HBM *Gochujang* group; DBM, HD with 10% of DBM *Gochujang* group; TG, triglyceride; TC, total cholesterol.

## Data Availability

Data is contained within the article or Appendix A.

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
