# Peer review of "Gochujang Ameliorates Hepatic Inflammation by Improving Dysbiosis of Gut Microbiota in High-Fat Diet-Induced Obese Mice"

_microorganisms, 2023, doi:10.3390/microorganisms11040911_

Round 1

Reviewer 1 Report

The paper submitted by Lee et al. is an interesting and well written article that shows that Gochujang fermented condiment ameliorates liver inflammation by improving the dysbiosis of the intestinal microbiota in obese mice, regardless of the salt content.

Apart from some minor details to be specified, outlined below, the study represents a carefully planned and performed research, worthy of publication.

• Paragraph 2.6: it is necessary to add the dilutions of antibodies used for the Western Blot.

• The meaning of c and d letters, as has been done for a and b, should be added to the captions of the figures 2,3,4,5,6,8 and table 2,3.

• Figure 4C: due to the poor resolution of the images, greater enlargement of a part of each image should be added in a square inside the current images.

• Figure 4D: How was the quantification of the liver lipid content assessed? The description must be added to the paragraph of Materials and Methods.

Reviewer 2 Report

Lee et al  fed B6 mice with five different diets (ND, HD, SALT, HBM and DBM) for 13 weeks and then performed analyses to assess inflammatory responses in various tissues and organs. The study is of interest and authors may wish to revise the manuscript addressing some issues and concerns:

Specific comments:

1.    Line 92-93 stated that blood and tissues were collected and stored at -80. Line 97-99 stated that TG, HDL, AST and ALT in serum were measured using commercial kits. This is confusing. Authors should stated clearly whether serum was separated from blood and then stored at -80 (should be) or whether blood was stored at -80 and then deforested and used to separate serum (as current statement imply).  

2.    Lines 137-147. There was no method description about gut microbiota analysis and no description of specific statistical software/program/method used for microbiota data analyses. Authors need to provide key methods, including key references of previously published methods, regarding microbiota analyses and related data analyses.

3.    Serval issues with data presented in Figure 2. Panel 1: remove all the statistical superscripts so that the curves could be clearly seen.  Panel B, C, D are based on food intake which is not necessary a reliable measurement unless authors housed mouse individually and specifically measured the amount of food wasted into bedding. Panel E: how could the SALT and BDM groups share a statistical superscript (no statistical difference) yet  have a star showing they are significantly different? Very much confusing and misleading.

4.    Figure 6F looks good but the data is hard to interpret. A 3-D plot could be better to show the data. Or authors could make an effort to show gut microbiota in a heatmap format at species (or other taxonomic) level with the X-axis showing all 35 animals (into 5 treatment groups) and the Y-axis showing an array of microbial species (top 30-50).  This type of heatmap should give a very good display of overall microbiota differences between different treatment groups. Data shown in Figure 7A and Figure 8A are very good!

5.    Discussions between lines 370 and 375 dealt with data presented in Figure 8A showing decline in Lactobacillus in both HBM and DBM groups relative to those in HD and SALT groups. With the data the authors have, why microbiota analysis was only performed at the genus level, but not at the species level, to address this issue directly? If data quality is not good enough to reveal microbiota differences at the species level, then authors should point that out specifically. While authors speculated on variable differences at the species and strain levels, the argument is not convincing since the differences at genus level is very large in the opposite direction. Notably Lactobacillus levels for HBM and DBM groups were relatively similar to that in the ND group (Figure 8A, 8C). Could that be a better argument for the discussion?

Reviewer 3 Report

The article discusses a very interesting and important topic which is the action of the traditional spice - Gochujang. This spice has anti-hepatitis anti-inflammatory properties, but often its beneficial effects are diluted by its high salt content. The authors of the article examined the effect of Gochujang as a means of preventing hepatitis and the composition of the intestinal microflora. The research material consisted of mice, which were divided into several research groups and a control group. Studies have shown that Gochujang significantly reduces lipid accumulation, liver damage, inflammatory response, and reduces the expression of a protein involved in the JNK/IκB/NF-κB pathway. Gochujang regulated the composition of the intestinal microflora.

Gochujang has demonstrated anti-inflammatory effects in the liver by reducing lipid accumulation, liver damage, and inflammatory response along with reorganization of intestinal microflora dysbiosis, independent of salt content and differences in microbial composition.

The research material consisted of mice properly prepared for the experiment. Statistical analysis was performed using SPSS using one-way ANOVA followed by Duncan's multiple range test. In the case of pairwise comparisons, the t-test was used. The statistical analysis was carried out correctly and indicates statistically significant differences between the groups.

Chart Figure 2A is hardly legible. I suggest changing the chart to a bar chart.

References contain up-to-date articles on the topic under study.
